

# Analytical performance evaluation of a multiplex real-time RT-PCR kit for simultaneous detection of SARS-CoV-2, influenza A/B, and RSV

Chi-Sheng Tai[1,*], Ming-Jr Jian[2,*], Tai-Han Lin[2], Hsing-Yi Chung[1,2], Chih-Kai Chang[2], Cherng-Lih Perng[2], Po-Shiuan Hsieh[1] and Hung-Sheng Shang[2]

[1] Graduate Institute of Medical Science, National Defense Medical Center, Taipei City, Taiwan
[2] Division of Clinical Pathology, Department of Pathology, Tri-Service General Hospital, National Defense Medical Center, Taipei city, Taiwan
[*] These authors contributed equally to this work.

Corresponding authors
Po-Shiuan Hsieh,
Pshsieh0465@gmail.com
Hung-Sheng Shang,
iamkeith@mail.ndmctsgh.edu.tw

## ABSTRACT

Differentiating between influenza, respiratory syncytial virus (RSV), and SARS-CoV-2 based on clinical symptoms alone can be challenging due to their overlap. This cross-sectional study, conducted from September to November 2023, evaluated the analytic performance of the LabTurbo multiplex real-time reverse transcription polymerase chain reaction (RT-PCR) kit for the simultaneous detection of these viruses. Nasopharyngeal swab specimens were tested using the LabTurbo kit, the Cobas Liat SARS-CoV-2 influenza A/B assay, and the Cobas influenza A/B and RSV assays. RNA standards were serially diluted and tested with the LabTurbo kit to determine the limit of detection (LOD). This cross-sectional study involved the analysis of 350 nasopharyngeal swab samples, which included 250 positive cases (50 cases each of influenza A, influenza B, and RSV, along with 100 cases of SARS-CoV-2) and 100 negative cases. The LabTurbo kit demonstrated 100% positive percent agreement and negative percent agreement with the reference assays for detecting SARS-CoV-2, influenza A/B, and RSV. The LODs for SARS-CoV-2, influenza A, influenza B, and RSV were 8,333, 3,333, 6,667, and 8,333 copies/mL, respectively. These findings confirm the diagnostic accuracy and analytic performance of the LabTurbo multiplex real-time RT-PCR kit for the detection of SARS-CoV-2, influenza A/B, and RSV, simultaneously. This assay could substantially improve the rapid identification and differentiation of these pathogens, thereby enabling more timely and appropriate treatment measures to control the spread of co-circulating viruses.

## INTRODUCTION

The COVID-19 pandemic not only brought global attention to the dangers of SARS-CoV-2 but also underscored the ongoing and significant threat posed by other respiratory viruses, especially influenza and respiratory syncytial virus (RSV), which continue to challenge public health systems worldwide (*Chow, Uyeki & Chu, 2023*; *Hirotsu et al.,*

*2024*; *Lai, Wang & Hsueh, 2020*; *Maltezou et al., 2023*). Consequently, various multiplex reverse transcription polymerase chain reaction (RT-PCR) assays have been developed and evaluated for the simultaneous detection of SARS-CoV-2, influenza viruses, and RSV (*Yun et al., 2021*; *Leung et al., 2021*; *Kim et al., 2022*). Recent studies have highlighted the clinical utility and performance of such multiplex panels. Each of these pathogens plays a substantial role in the global burden of respiratory illnesses, with profound socioeconomic impacts and challenging implications for public health. Influenza viruses, with their rapid mutation rates and annual cyclical patterns, pose perennial global health challenges (*Shao et al., 2017*). The ability of these viruses to cause annual epidemics and occasional pandemics underscores their threat to human health. Clinically, influenza can range from mild to severe, with high-risk populations, including the elderly, young children, and individuals with underlying health conditions, facing the greatest risk for severe outcomes (*Adlhoch et al., 2019*; *Famati et al., 2023*). Epidemiologically, the global burden of influenza is staggering, with an estimated three to five million cases of severe illness and up to 650,000 respiratory deaths annually (*Famati et al., 2023*). The economic impact is equally substantial, encompassing healthcare costs, lost productivity, and strain on healthcare resources during the peak flu seasons. RSV is another major contributor to respiratory infections, particularly in infants and elderly individuals. Unlike influenza, RSV has not garnered the same level of public attention, yet it is the leading cause of bronchiolitis and pneumonia in children under one year of age globally (*Baraldi et al., 2022*). In older adults, RSV can lead to serious respiratory diseases, mirroring the clinical impact of influenza in this population. The virus exhibits marked seasonality in temperate climates, with yearly outbreaks causing substantial hospitalization rates among infants and increased mortality rates among the elderly and those with chronic illnesses (*Cocchio et al., 2023*). The recent availability of three RSV vaccines has significantly improved prevention efforts; however, challenges in controlling and managing RSV infections persist. The COVID-19 pandemic has further complicated the epidemiological landscape, demonstrating the simultaneous circulation and interaction of SARS-CoV-2, influenza, and RSV. The overlapping clinical presentations of these viruses pose substantial challenges for diagnosis and treatment, and often require sophisticated diagnostic tools to accurately differentiate between them (*Dadashi et al., 2021*). This convergence also highlights the potential for coinfections, which can exacerbate clinical outcomes and complicate therapeutic strategies. As the pandemic progresses, the need for vigilant surveillance, effective vaccines, and rapid diagnostic technologies becomes increasingly apparent to mitigate the threats posed by these respiratory viruses and to safeguard public health (*Cilloniz et al., 2022*).

A previous study discovered that social distancing measures implemented during the COVID-19 pandemic considerably reduced the transmission of other respiratory pathogens, such as RSV and influenza in a community in Brazil. During the autumn and winter of 2020, in a study involving individuals seeking care for COVID-19-like symptoms, SARS-CoV-2 was detected in 32.7% of the participants, whereas influenza and RSV were not detected (*Varela et al., 2021*). This suggests that preventive social measures aimed at reducing the spread of SARS-CoV-2 also considerably affect the transmission of other respiratory pathogens (*Du et al., 2021*; *Sarvepalli et al., 2021*; *Varela et al., 2021*).

As social interactions return to pre-2020 levels, with schools fully open and office spaces buzzing again, familiar seasonal illnesses reappear. Viruses like the flu and RSV, which had taken a backseat during the period of limited contact, are finding more opportunities to spread as people reconnect. This shift brings a renewed focus on protecting community health. Seasonal vaccines play a crucial role in preventing these common viruses from overwhelming our healthcare systems. Equally important are the continued efforts to track disease patterns and educate the public about effective prevention strategies–from proper handwashing to covering coughs and sneezes. By staying vigilant and maintaining these healthy habits, communities can better manage the return of these everyday health challenges (*Du et al., 2021*; *Varela et al., 2021*).

Testing for multiple respiratory infections at once has become essential as viruses can strike together. The traditional workflows involving sequential, single-target RT-PCR assays or viral culture, which require more hands-on time, manual processing, and longer turnaround (often over 6 h for RT-PCR or several days for culture), as compared to modern multiplexed, automated molecular platforms. Better testing tools that can spot different viruses quickly and accurately are now needed. Finding out what is making someone sick early on helps doctors provide the right treatment and helps health officials prevent these illnesses from spreading further in the community (*Pandey et al., 2022*; *Safiabadi Tali et al., 2021*).

Historically, Taiwan has experienced annual fluctuations in the prevalence of respiratory viruses, with distinct peaks in influenza and RSV infections corresponding to seasonal variations (*Hsu et al., 2022*; *Lin et al., 2022*). The COVID-19 pandemic introduced an additional layer of complexity to this epidemiological landscape, with the emergence of SARS-CoV-2 leading to overlapping epidemics. This convergence underscores the necessity for robust diagnostic tools capable of swiftly distinguishing between these pathogens to effectively guide clinical and public health responses.

Multiplex assays offer advantages over traditional single-target tests by streamlining diagnostics and reducing the time and resources needed to identify pathogens. The timely identification of the causative respiratory pathogen is crucial for guiding appropriate treatment. For example, patients confirmed to have influenza may benefit from neuraminidase inhibitors (*e.g.*, oseltamivir), while those with SARS-CoV-2 infection may be eligible for targeted antivirals such as remdesivir or nirmatrelvir/ritonavir. In contrast, accurate identification of RSV, especially in pediatric or geriatric patients, helps prompt supportive care and targeted infection control. Rapid, specific diagnosis also helps reduce unnecessary antibiotic use and informs public health isolation or surveillance strategies. This expedited testing is crucial in acute care settings, where prompt results can directly impact treatment decisions, potentially enhancing patient outcomes and mitigating disease spread. The high sensitivity and specificity of these methods further bolster their reliability, making them valuable tools in clinical and public health spheres.

This research aimed to evaluate the LabTurbo multiplex real-time RT-PCR kit for the simultaneous detection of SARS-CoV-2, influenza A/B, and RSV. We assessed the kit's diagnostic performance using 350 clinical specimens, comparing results with established methods (Cobas SARS-CoV-2 & influenza A/B, and Cobas influenza A/B & RSV assays on

the Cobas Liat system). This evaluation sought to determine the kit's efficacy and reliability in clinical settings. The assessment is crucial for validating the kit as an effective tool to optimize diagnostics, thereby enhancing capabilities for swift therapeutic interventions and informed public health decision-making.

## MATERIALS AND METHODS

### Study design and clinical specimens

This cross-sectional study was conducted in a contracted laboratory affiliated with the Tri-Service General Hospital in Northern Taiwan, covering the densely populated areas of Taipei City and New Taipei City with an approximate combined population of 6.6 million. The investigation took place between September and November 2023. We conducted a parallel comparison involving the analysis of historically processed nasopharyngeal specimens collected in a universal transport medium. Specimens were selected not only based on the availability of an associated electronic record but also with the intent to include a balanced representation of respiratory viruses. Specifically, 50 cases each of influenza A, influenza B, and RSV, along with 100 cases of SARS-CoV-2, were intentionally selected to ensure a comprehensive evaluation of the assay's performance across these key pathogens. Inclusion criteria encompassed patients of all ages who presented with symptoms of upper respiratory tract infections, such as fever, cough, and sore throat. We also identified and reported the specific SARS-CoV-2 variants circulating during the study period to provide a more detailed context for the results. The nasopharyngeal swabs were collected in a universal transport medium (Libo, New Taipei city, Taiwan) and stored at −80 °C until analysis. The duration of sample storage before evaluation was carefully documented to ensure the integrity of the specimens during the diagnostic process. The study design is schematically represented in Fig. 1.

This study was approved by the Institutional Review Board of the Tri-Service General Hospital (approval number B202305091; approval date July 19, 2023). Informed consent was obtained from all the participants, highlighting our contracted laboratory's dedication to providing valuable insights into multiple upper-pathogen diagnostic advancements through rigorous research practices.

### Viruses detection platform/system

In this study, we utilized the LabTurbo multiplex real-time RT-PCR kit (LabTurbo, New Taipei City, Taiwan) as the primary platform for virus detection, following the manufacturer's operation guidelines. This advanced kit enabled simultaneous and efficient detection of SARS-CoV-2, influenza A/B, and RSV. The LabTurbo system facilitated automated, high-throughput analysis, enhancing both the accuracy and efficiency of viral identification

### LabTurbo AIO COVID-19/FLU A/FLU B/RSV RNA testing

The LabTurbo AIO COVID-19/FLU A/FLU B/RSV RNA testing kit (LabTurbo) is a modern multiplex RT-PCR kit designed for efficient viral detection engineered for the quantitative identification of nucleic acids from influenza A (subtypes H1, H1N1, and H3), influenza

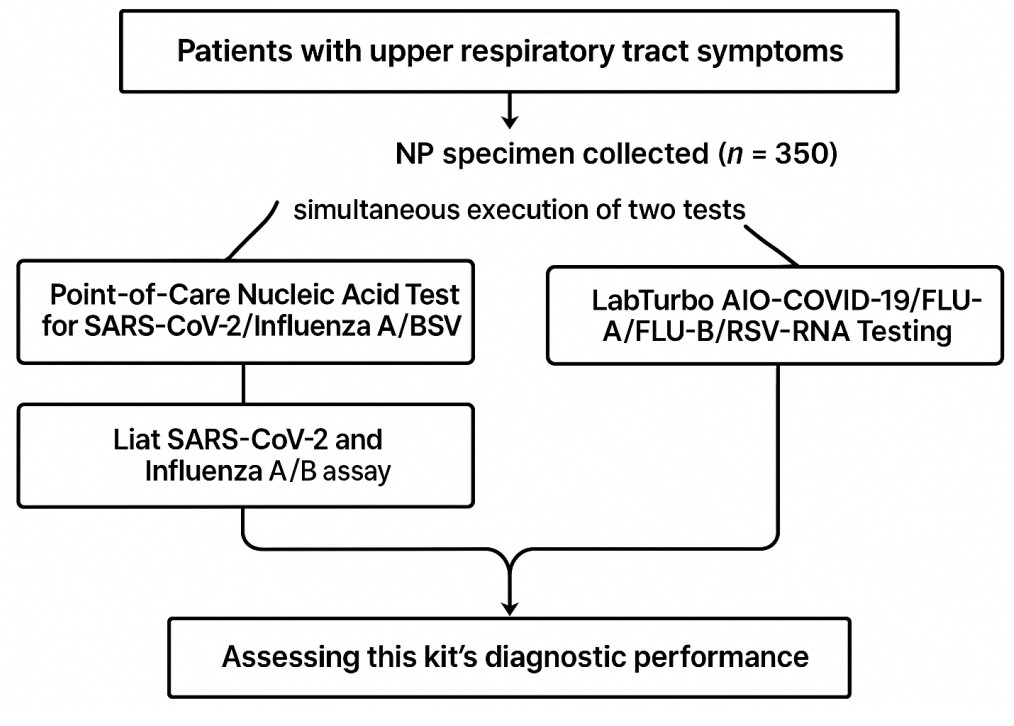

**Figure 1** Flow chart in this study design.

B, RSV, and SARS-CoV-2 in upper respiratory specimens such as nasopharyngeal swabs. This comprehensive kit enables the concurrent detection of specific genetic markers for influenza A ($M$ gene), influenza B ($M$ gene), and RSV ($N$ gene), and incorporates the detection of the SARS-CoV-2 target gene ($RdRP$ gene).

LabTurbo AIO system utilizes a fully automated workflow for nucleic acid extraction and real-time PCR (qPCR) detection, substantially reducing manual handling and processing time including: 1. Automated Workflow Efficiency: The LabTurbo QuadAIO system processes up to 96 samples per run, with a hands-on time of less than 10 min per batch. The system performs all steps, from sample lysis, nucleic acid extraction, to qPCR setup and detection, automatically. LabTurbo AIO seamlessly integrates Peltier thermoelectric elements with PCR instrumentation for precise temperature control. This process includes sample insertion, extraction, reaction setup, qPCR analysis, and automatic generation of cycle threshold (Ct) reports, with the capacity to analyze up to 1,200 samples daily, thus providing Ct reports directly for clinical specimen testing, as described previously (*Jian et al., 2021*).

## RT-qPCR reagent preparation

The LabTurbo diagnostic tests target specific genes for each pathogen. The $RdRP$ gene is targeted for SARS-CoV-2, the $M$ gene for influenza A and B, and the $N$ gene for RSV. Primers and probes were designed to specifically amplify these target regions, ensuring high specificity and sensitivity. The preparation involved assembling the necessary components,

including Reverse Transcriptase plus (RT+), 2X PCR Master Mix (MM), primer/probe mixture CFR (PM CFR), and RNase-free water. A standard PCR mix with the following components: 1.25 μL of Reverse Transcriptase Plus (RT+) for transcribing RNA into DNA, 12.5 μL of 2X PCR Master Mix for the necessary enzymes and buffers, 2.5 μL of the primer/probe mixture for specific DNA binding, 2.75 μL of RNase-free water to prevent unwanted RNase activity, and 7.5 μL of RNA template with internal controls for genetic material amplification. A distinct feature of the LabTurbo AIO system is its swift turnaround time, offering results in as little as 2.5 h. It requires minimal manual sample preparation (approximately 1 min) and supports high throughput results.

The PCR amplification was conducted utilizing the following thermocycling parameters: initiation of the reverse transcription reaction (RT reaction) was set at 55 °C for 480 s, followed by a solitary cycle. Heat activation of the reaction was performed at 95 °C for 60 s, limited to one cycle. Subsequent denaturation was maintained at 95 °C for 1 s, for 45 cycles. Annealing and extension phases were performed at 60 °C for 15 s each, spanning 45 cycles. The thermal profile was tailored for efficient amplification and monitored in real time using a quantitative fluorescent assay for the precise and immediate detection of DNA amplification products.

## Assessment of analytical sensitivity and specificity

To evaluate the sensitivity and specificity of our designed multiplex assay, a series of tests were conducted to confirm its ability to detect a wide range of pathogens, including SARS-CoV-2, influenza A (subtypes H1, H1N1, and H3), influenza B, and RSV (subtypes A and B). Multiplex RT-PCR was performed on the LabTurbo AIO system using AmpliRun controls, obtained from Launch Diagnostics to assess the limit of detection (LOD) for the multiplex RT-PCR assay. For assay validation and LOD determination, we utilized the AMPLIRUN® SARS-CoV-2 RNA Control, as well as corresponding controls for influenza A/B and RSV. These controls are purified nucleic acid standards specifically designed for use in nucleic acid amplification-based techniques and are suitable for any molecular testing platform. Each control consists of purified nucleic acid derived from the complete microbial genome, allowing amplification of any relevant target. The controls are provided at precise concentrations, quantified by qPCR, typically within a range of 12,500–20,000 copies/μL, as specified by the manufacturer. These controls provided precise viral RNA concentrations for SARS-CoV-2, influenza A/B, and RSV, allowing for accurate determination of the assay's sensitivity with known concentrations provided by the manufacturer. The controls were used to prepare serial dilutions, ensuring that the assay's performance was rigorously validated across a range of viral loads. This process aimed to determine the LOD by conducting the experiment 20 times. The LOD was defined as the lowest detectable concentration with a positive detection rate of 95%. The analytical sensitivity of the LabTurbo AIO test was determined as the lowest dilution at which all replicates were identified as positive. The LOD values presented were determined through a series of two-fold dilutions of RNA standards. This approach ensured precise and accurate LOD determination. The values were not arbitrarily rounded but were based on the lowest concentration at which 95% of the replicates were consistently detected.

## Point-of-care nucleic acid testing for SARS-CoV-2/influenza A/B RSV

In cases of upper respiratory tract infections, our diagnostic strategy incorporated the use of the Liat SARS-CoV-2 and influenza A/B assays (Roche Molecular Systems, Pleasanton, CA, USA) and Liat influenza A/B & RSV test (Roche Molecular Systems) executed on the Liat analyzer. This pivotal assay forms an integral part of our point-of-care nucleic acid amplification testing platform, offering streamlined, automated, rapid sample processing through real-time RT-PCR, with results available in less than 30 min. Their multiplex functionality facilitates the simultaneous identification of SARS-CoV-2, influenza A, and influenza B, aiding in the differential diagnosis of such infections. The Roche test systems target the following genes: SARS-CoV-2 ($E$ and $N$ genes), influenza A ($M$ gene), influenza B ($NS$ gene), and RSV ($N$ gene). This study designates the method employed by the LIAT assay as the standard technique for diagnostic evaluations and comparison with the LabTurbo multiplex nucleic acid diagnostic kit. In this study, the LOD was experimentally determined only for the LabTurbo multiplex real-time RT-PCR kit. The Cobas Liat system was used as a reference comparator for clinical performance metrics (PPA/NPA), consistent with its established use in clinical laboratories. We did not independently determine the LOD for the Cobas Liat assays, and our comparison between platforms is therefore based on clinical concordance rather than analytical sensitivity.

## Virus culture

Virus culture was conducted using MDCK cells (*Canis familiaris*, dog kidney, female, epithelial, CCRC 60004; ATCC CCL-34) and A549 cells (human lung carcinoma, male, Caucasian, epithelial, ATCC CCL-185). The cells were seeded in culture vessels and grown to 70–90% confluence. MDCK cells were maintained in Dulbecco's Modified Eagle Medium supplemented with 10% fetal bovine serum and antibiotics (penicillin 10,000 units/mL, streptomycin 10 mg/mL, amphotericin B 0.025 mg/mL). A549 cells were cultured in Minimum Essential Medium with 10% fetal bovine serum and the same antibiotic combination. The viruses, including strains of influenza A, influenza B, and RSV, originated from clinical specimens collected from patients in this study.

During the cultivation, cell morphology was closely monitored for cytopathic effects, such as cell rounding, detachment, and syncytium formation. Media were refreshed every 24–48 h, and supernatants were collected for further analysis. Once significant cytopathic effects was observed, the viral supernatant was harvested, centrifuged at 3,000× g for 10 min at 4 °C to remove cellular debris, aliquoted, and stored at −80 °C for subsequent experiments.

## Statistical analyses

Data statistical analysis utilized Excel (Microsoft Corp., Redmond, WA, USA) and GraphPad Prism Version 8.0 (GraphPad, Inc., San Diego, CA, USA) for thorough examination and visualization of results.

The study employed positive percent agreement (PPA) and negative percent agreement (NPA) with confidence intervals (CI), alongside correlation analysis between quantification cycle (Cq) values from the LabTurbo and Liat assays. Statistical methods were selected

**Table 1  Clinical performance of the LabTurbo AIO COVID-19/FLU A/FLU B/RSV RNA testing.**

| Liat result | | LabTurbo Quadruplex result | | PPA[*] (95% CI[*]) | NPA[*] (95% CI) |
|---|---|---|---|---|---|
| | | Positive | Negative | | |
| SARS-CoV-2 | Positive | 100 | 0 | 100% (96.3–100%) | 100% (96.3–100%) |
| | Negative | 0 | 100 | | |
| Influenza A | Positive | 50 | 0 | 100% (92.9–100%) | 100% (96.3–100%) |
| | Negative | 0 | 100 | | |
| Influenza B | Positive | 50 | 0 | 100% (92.9–100%) | 100% (96.3–100%) |
| | Negative | 0 | 100 | | |
| RSV | Positive | 50 | 0 | 100% (92.9–100%) | 100% (96.3–100%) |
| | Negative | 0 | 100 | | |

**Notes.**
 *PPA, positive percent agreement; NPA, negative percent agreement; CI, confidence interval.

to align with the data characteristics and research objectives, focusing on evaluating and comparing the effectiveness of various diagnostic tests.

## RESULTS

### Evaluation of LabTurbo multiplex nucleic acid diagnostic kit for respiratory pathogens

This study meticulously examined 350 samples from 350 different patients, including 100 negative and 250 positive specimens. In this study, specimens were collected from 350 patients exhibiting the following characteristics: the age range spanned from infants to elderly adults, with a gender distribution of 180 males and 170 females. These patients primarily presented with symptoms of upper respiratory tract infections, including fever, cough, and sore throat. These general characteristics of the patient cohort provide valuable clinical context, forming the foundation for further research and analysis of the etiology of upper respiratory tract infections and their manifestations across different age groups and genders. The positive cases comprised 50 cases of influenza A, 50 cases of influenza B, 50 cases of RSV, and 100 cases of SARS-CoV-2. Three cases of co-infection were identified within this dataset, enriching the complexity of our analysis. These included one case of concurrent infection with COVID-19 and influenza A and two cases of co-infection involving influenza A and RSV.

This analysis utilized an advanced multiplex nucleic acid diagnostic kit developed by LabTurbo Biotechnology Co., Ltd. to identify the presence of SARS-CoV-2, influenza A, influenza B, and RSV. These positive results were subsequently compared to the benchmark set by the Roche Cobas influenza A/B and RSV test kits/SARS-CoV-2 and influenza A/B assay, utilized as the standard method for this study, with all evaluations conducted by skilled medical laboratory technicians. Both the Cobas and LabTurbo methods yielded identical results for detecting SARS-CoV-2, influenza A, influenza B, and RSV in negative and positive samples, with no reported discrepancies, and the calculations for PPA and NPA showed perfect agreement (100%) (Table 1).

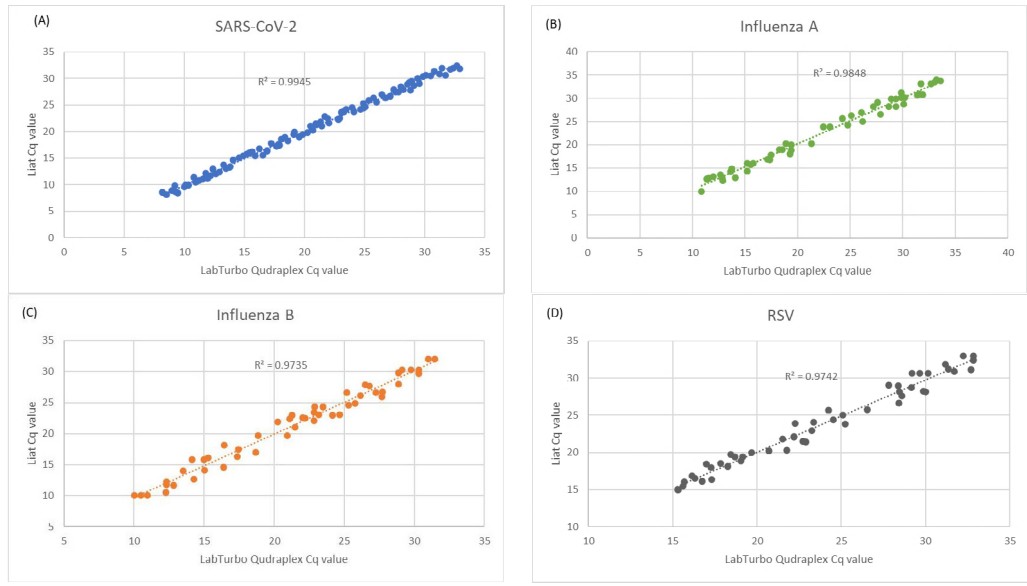

**Figure 2** Correlation between Cq values of clinical specimens tested positive by both the Liat and LabTurbo Quadruplex assays.

The Cq values of the clinical specimens tested and analyzed using both the Liat and LabTurbo Quadruplex assays for SARS-CoV-2, influenza A, influenza B, and RSV demonstrated a high correlation, with $R^2$ values ranging from 0.9335 to 0.9945 (Fig. 2). This finding highlights the precision and consistency of the Liat and LabTurbo Quadruplex assays for measuring Cq values across viral pathogens and affirms their reliability for comprehensive respiratory virus diagnostics.

## Analytic sensitivity of the LabTurbo multiplex nucleic acid diagnostic kit detecting SARS-CoV-2, influenza A/B, and RSV

The detection limits for SARS-CoV-2, influenza A/B, and RSV were established using the AMPLIRUN controls for SARS-CoV-2, influenza A H1/H3/H1N1, influenza B, and RSV (subtypes A/B). These controls, provided by Vircell, comprised purified RNA from the respective viral genomes, facilitating precise quantification. LOD studies determine the lowest detectable concentration of SARS-CoV-2 at which greater or equal to 95% of all (true positive) replicates that tested positive. The data provided specify the LOD for various respiratory pathogens, measured in copies per milliliter. SARS-CoV-2 was detectable at concentrations as low as 8,333 copies per milliliter, demonstrating the assay's adequate sensitivity. In contrast, influenza A had a difference in LOD values and assay performance, detectable at only 3,333 copies per milliliter, whereas influenza B required a slightly higher concentration of 6,667 copies per milliliter, indicating different strain sensitivities or assay variations. RSV could be detected at 8,333 copies per milliliter, comparable to that of SARS-CoV-2, highlighting the capacity of the assay to effectively identify early or low-level infections (Table 2). Also, to further verify the reliability of our multiplex testing method, we conducted additional tests on mixed samples. These tests involved combining RNA

**Table 2   Limit of detection (LOD) for LabTurbo multiplex nucleic acid diagnostic kit assays.**

| | Analytical sensitivity |
|---|---|
| **Pathogen** | **LOD per milliliter** |
| SARS-CoV-2 | 8,333 |
| Influenza A | 3,333 |
| Influenza B | 6,667 |
| RSV | 8,333 |

**Table 3   Analytical specificity of the LabTurbo multiplex nucleic acid diagnostic kit.**

| | LabTurbo multiplex nucleic acid diagnostic kit | | | |
|---|---|---|---|---|
| | **SARS-CoV-2** | **Influenza A** | **Influenza B** | **RSV** |
| **Clinical viral isolated with known viruses** | | | | |
| Rhinovirus ($n = 25$) | Not detected | Not detected | Not detected | Not detected |
| Adenovirus($n = 25$) | Not detected | Not detected | Not detected | Not detected |
| Parainfluenza 1 virus ($n = 25$) | Not detected | Not detected | Not detected | Not detected |
| Parainfluenza 2 virus ($n = 25$) | Not detected | Not detected | Not detected | Not detected |
| Parainfluenza 3 virus ($n = 25$) | Not detected | Not detected | Not detected | Not detected |

from different viruses in various concentrations and testing them using the LabTurbo AIO system. The results confirm the system's capability to accurately detect multiple pathogens in a single assay.

In this study, the analytical specificity was appraised for a laboratory-developed dual multiplex PCR assay using the LabTurbo multiplex nucleic acid diagnostic kit and samples known to contain upper respiratory viruses, including rhinovirus, parainfluenza virus, and adenovirus. The evaluation was further extended to include undiluted supernatants derived from cell cultures. The test consistently exhibited high specificity for predetermined viral targets. No evidence of cross-reactivity with other upper respiratory viruses was observed, as substantiated by the data delineated in Table 3.

## Comparison of viral culture and PCR testing outcomes

We examined the discrepancies between viral cultures and PCR testing outcomes for various respiratory viruses. We focused on three specimens that showed discordant results between the two diagnostic methods, highlighting the challenges in viral diagnostics (Table 4). For specimen number 117, the presence of influenza A (Ct 17.62) and RSV (Ct 29.38) was determined using PCR. In contrast, the viral culture showed growth only for influenza A. The higher sensitivity of PCR may have detected RSV at lower levels that were not viable for culture, or there may have been a selective growth preference for influenza A due to the culture media used. Similarly, for specimen number 242, the PCR assay detected influenza A (Ct = 17.62) and RSV (Ct = 29.38); however, the culture process only yielded RSV.

**Table 4  Details of three specimens showing discordant results between PCR and viral culture.**

| Specimen no. | Clinical discrepancies | |
|---|---|---|
| | LabTurbo quadruplex assay | Culture |
| 65 | Influenza A (27.82), SARS-CoV-2 (31.13) | Influenza A only[*] |
| 117 | Influenza A (17.62), RSV (29.38) | Influenza A only |
| 242 | Influenza A (19.68), RSV (27.68) | RSV only |

**Notes.**
*The cultivation of SARS-CoV-2 falls outside the scope of our contracted laboratory's capabilities.

## DISCUSSION

In the context of the ongoing COVID-19 pandemic, rapid and accurate differential diagnosis of respiratory pathogens, including SARS-CoV-2, influenza A and B, and RSV, is important for patient management, public health surveillance, and containment strategies. This study evaluated and compared the performance and analytical performance of the LabTurbo AIO system-based molecular diagnostic methods for the rapid identification of these respiratory viruses. The LabTurbo AIO platform represents a substantial advancement in the field of molecular diagnostics, offering rapid, sensitive, and specific detection of common respiratory viruses using real-time RT-PCR assays, which are the gold standard for viral nucleic acid detection because of their high sensitivity and specificity.

The discrepancies between viral cultures and PCR testing outcomes could suggest competitive inhibition, where one virus outgrows the other in culture, or it may indicate a lower infectious titer of influenza A under culture conditions. These results underscore the complexity of interpreting diagnostic tests for respiratory viruses, where PCR offers broader detection capabilities that may not always correlate with growth potential in viral cultures. Understanding the implications of these discordant findings is important for clinical decision-making and public health responses, especially in managing co-infections. These data also emphasize the need to consider both Cq values and clinical symptoms to guide treatment strategies.

Previous studies have revealed the application and effectiveness of PCR testing techniques for the simultaneous detection of SARS-CoV-2, influenza A/B, and RSV (*Leung et al., 2021*; *Yun et al., 2021*). Yun et al., evaluated three commercially available multiplex real-time reverse transcription PCR kits, which showed high sensitivity and specificity for detecting targeted viruses without cross-reactivity. Notably, the Allplex kit showed superior sensitivity for RSV detection. *Leung et al. (2021)* assessed the Xpert Xpress SARS-CoV-2/Flu/RSV kit (Xpert 4-in-1), highlighting its rapid and reliable detection capabilities, with over 99.64% concordance with comparator assays, offering a considerable advantage in emergency testing scenarios. *Kim et al. (2022)* compared the Kaira COVID-19/Flu/RSV detection kit with the PowerChek kit, and both demonstrated comparable performance in virus detection (*Kim et al., 2022*).

Literature comparing multiplex and singleplex RT-PCR assays for respiratory viruses has yielded varied results. Some studies suggest that well-optimized multiplex assays can achieve analytical sensitivity comparable to, or even enhanced compared to, their singleplex counterparts for clinically significant viruses (*Feng et al., 2024*; *Thieulent et al., 2024*). For

example, *Feng et al. (2024)* pointed out that for certain virus/assay combinations, or when viral loads are very low, singleplex assays might offer marginally lower detection limits.

To provide a broader clinical context for the application of the LabTurbo system, it is important to distinguish its role from other commonly used diagnostic modalities, particularly rapid antigen tests. While not directly evaluated in this study, these alternative approaches serve different purposes and are based on fundamentally different technologies. Lateral flow assays (rapid antigen tests) represent a different category of diagnostic tools, which typically offer faster turnaround times (*e.g.*, 15–30 min) but generally have lower sensitivity compared to NAATs like RT-PCR. While a direct comparison was not part of this study, this discussion acknowledges that RT-PCR assays like LabTurbo are generally indicated where higher sensitivity is required, such as in hospitalized patients or for confirmatory testing, whereas rapid antigen tests might be used for broader, more rapid screening in community settings. Their use scenarios and significance differ based on these performance characteristics.

This study further extends the evaluation of the multiplex RT-PCR assay by focusing on its diagnostic performance against the predominant SARS-CoV-2 variants circulating in Taiwan from September to December 2023, specifically JN.1, KP.2, and KP.3. These variants exhibit unique transmission dynamics and immune evasion properties, which could potentially impact the sensitivity and specificity of existing diagnostic tools.. However, the current study adds significant value by addressing the assay's effectiveness in the face of newer variants and during simultaneous outbreaks of SARS-CoV-2, influenza A/B, and RSV. The ability to rapidly differentiate between these co-circulating pathogens is critical as we transition from pandemic to endemic phases, where timely and accurate diagnostics are necessary for effective patient management and public health strategies. These insights underscore the assay's ongoing utility and the importance of continuous validation against emerging viral threats.

Additionally, we have elaborated on the advantages of the LabTurbo system in comparison to traditional RT-PCR platforms. The LabTurbo AIO platform enables automated processing of up to 96 samples per run with minimal hands-on time ($\sim$10 min), and a full sample-to-result turnaround of approximately 2.5 h. This is significantly faster than conventional RT-PCR workflows, which typically take 4–6 h due to manual extraction and preparation steps. The system's high throughput capacity (up to 1,200 samples/day) makes it suitable for use in centralized clinical laboratories or during large-scale screening campaigns.

We also acknowledge that the LabTurbo system may be less suitable for low-resource settings due to its capital equipment cost (approximately $90,000–120,000 USD) and need for skilled operation. The cost per test is approximately $18–22 USD. Therefore, its optimal use scenario is in high-volume testing environments rather than in decentralized or remote areas.

This study has several limitations. First, the cross-sectional design limits the ability to establish causality. Second, the sample size, although adequate for preliminary analysis, may not be large enough to detect less common co-infections or variations in viral load among different populations.

## CONCLUSIONS

This analysis highlights the LabTurbo quadruplex nucleic acid test kit as an effective tool for simultaneously detecting SARS-CoV-2, influenza A/B, and RSV. The sample-to-result automation offers significant advantages, including high throughput (up to 1,200 samples/day), minimal hands-on time, and rapid turnaround ($\sim$2.5 h), which are particularly beneficial for time-sensitive diagnostic needs in clinical laboratories. Moreover, the system demonstrated excellent diagnostic accuracy, with 100% PPA and NPA compared to reference assays. Although the Cobas Liat system is capable of delivering individual test results in under 30 minutes—making it highly suitable for urgent or point-of-care diagnostics—the LabTurbo system is designed for high-throughput operation, with automated workflows that are optimized for large batch processing in centralized laboratory settings.

However, we also recognize certain limitations. As with many multiplex assays, the sensitivity for individual targets may be slightly lower than that of dedicated single-target RT-PCR tests. While this trade-off is generally acceptable in typical clinical scenarios, it may be a consideration when ultra-sensitive detection of a specific virus (*e.g.*, early-stage SARS-CoV-2 infection) is required. Nevertheless, the ability to simultaneously detect multiple pathogens remains a crucial advantage, especially during respiratory virus season or outbreak conditions.

In summary, the LabTurbo system provides a robust, scalable, and clinically practical solution for respiratory virus detection, balancing speed, scope, and diagnostic accuracy.

### Funding

This work was supported by Tri-Service General Hospital, Taipei (grant numbers TSGH-D-113105 and TSGH-D-113106). The funders had no role in study design, data collection and analysis, decision to publish, or preparation of the manuscript.

### Grant Disclosures

The following grant information was disclosed by the authors:
Tri-Service General Hospital, Taipei: TSGH-D-113105 and TSGH-D-113106.

### Competing Interests

The authors declare there are no competing interests.

### Author Contributions

- Chi-Sheng Tai performed the experiments, analyzed the data, prepared figures and/or tables, and approved the final draft.
- Ming-Jr Jian performed the experiments, analyzed the data, prepared figures and/or tables, and approved the final draft.
- Tai-Han Lin performed the experiments, analyzed the data, prepared figures and/or tables, and approved the final draft.

- Hsing-Yi Chung performed the experiments, analyzed the data, prepared figures and/or tables, and approved the final draft.
- Chih-Kai Chang performed the experiments, analyzed the data, prepared figures and/or tables, and approved the final draft.
- Cherng-Lih Perng conceived and designed the experiments, performed the experiments, analyzed the data, prepared figures and/or tables, authored or reviewed drafts of the article, and approved the final draft.
- Po-Shiuan Hsieh conceived and designed the experiments, authored or reviewed drafts of the article, and approved the final draft.
- Hung-Sheng Shang conceived and designed the experiments, authored or reviewed drafts of the article, and approved the final draft.

### Human Ethics

The following information was supplied relating to ethical approvals (i.e., approving body and any reference numbers):

This study was approved by the Institutional Review Board of Tri-Service General Hospital (TSGHIRB No. B202305091, registered on July 19, 2023).

### Data Availability

The data is available in the Supplementary File.

### Supplemental Information

Supplemental information for this article can be found online at http://dx.doi.org/10.7717/peerj.19693#supplemental-information.

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
