# Peer review of "Analytical performance evaluation of a multiplex real-time RT-PCR kit for simultaneous detection of SARS-CoV-2, influenza A/B, and RSV"

_PeerJ, doi:10.7717/peerj.19693_

## Round 0.1 · original submission · Major Revisions

·

Basic reporting

The study describes the performance of a multiplex kit targeting SARS-COV-2, RSV, and Influenza A/B. Authors compares the performance of the kit with another multiplex kit. Although the study novelty is weak, the study can be useful for researchers and laboratories who work in molecular diagnosis.

The literature should be improved by including the most recent similar studies. The study need more discussion by referring to previous studies. Especially, the sensitivity of the multiplex assay should be compared with the other assays that target single viral genome, such as SARS-COV-2. It seems that the sensitivity of the kit is likely lower compared with the single virus targeting kits.

Experimental design

The LOD calculation should be explained clearly. Authors state that `These controls provided precise viral RNA concentrations for SARS CoV-2, Influenza A/B, and RSV`. How are the RNA concentrations calculated? Did the authors use synthetic controls or culture? How the RNA copy number is calculated?
Some typological and grammatical mistakes need to be fixed.
The abbreviations should only be explained at first.
The authors stated that Detailed information on the primers and probes used is provided in the Methods section. However, I couldn`t find it in the manuscript.
How about the LODs of the other kits tested?

Validity of the findings

The conclusion section should be expanded by indicating the pros and cons of the current tests.
The time and sensitivity comparison of the tested kits should also be shown.
Also, the sensitivity of the tested multiplex kits should be compared with other multiplex kits that target a single virus, such as SARS-CoV-2. This is important in selecting the correct kit for diagnosis.

Reviewer 2 ·

Basic reporting

This study demonstrated the accuracy and analytical performance of the LabTurbo Multiplex Real-Time RT-PCR Kit for simultaneous detection of SARS-CoV-2, influenza A/B, and RSV. The nasopharyngeal swab sample size was 350, including 250 positive samples (50 influenza A, 50 influenza B, 50 RSV, and 100 SARS-CoV-2) and 100 negative samples. They demonstrated 100% PPA and high sensitivity of the LabTurbo system.

Experimental design

The paper highlights the importance of testing for multiple respiratory infections simultaneously. However the advantages of LabTurbo system are not fully addressed. Here are some suggestions for further revisions.
1. What are the advantages of the LabTurbo system compared to traditional PCR in terms of response time and throughput?
2. Is the LabTurbo system suitable for low-resource settings, and what are the costs of the equipment and each test?
3. How does the performance of the LabTurbo system compare to the 6-in-1 lateral flow assay (rapid antigen test in the market), e.g. reaction time and sensitivity. Please identify their use scenarios and what their significance is.

Validity of the findings

no comment

Additional comments

no comment

·

Basic reporting

No comment.

Experimental design

While the experimental design is appropriate and the results are correctly interpreted, how relevant this work is to the field remains in question. After evaluating the manuscript, it is unclear what the advantage of this assay is compared to the many others that are widely available.

Validity of the findings

No comment

Additional comments

In this study, Tai et al. assessed the analytical performance of the LabTurbo Multiplex Real-time RT-PCR Kit for the simultaneous detection of Flu A, Flu B, RSV, and SARS-CoV-2 by conducting a cross-sectional comparison to multiple standard assays currently in use. They observed 100% PPA and NPA, which would also indicate perfect sensitivity and specificity. Additionally quantification measures were similar between both assays as measured by high correlation values for Cqs produced from matched samples. The authors provide clear clinical and public health rationale for the ability to rapidly distinguish between different respiratory pathogens that present clinically with similar symptomology. The methodology for the study is appropriate, as is the analytical approach. Taken together, this paper represents a thorough evaluation of the analytical performance of the LabTurbo Multiplex RT PCR kit. However, what is not addressed in this paper are the following: how does this assay perform considering substantial genetic diversity present in influenza, SARS-CoV-2, and RSV? There is discussion of primer sets included in the assay, however there is not discussion of how these are monitored and updated in light of circulating genetic diversity. Which begs the next point- what is the added benefit of the LabTurbo Multiplex Realtime RT-PCR Kit compared to the numerous other nucleic acid based diagnostics on the market for the simultaneous detection of multiple respiratory pathogens? This is all to say that the paper is nicely put together and the authors thoroughly demonstrate the benefit of this assay, but what remains unclear is what the added benefit of the assay is?
Minor Comments:
Line 97- the authors state that the “old way” for identifying viruses is too slow without describing what the “old way” is. I would suggest the authors better frame this work in the context of current diagnostic standards for respiratory viruses.
Line 99- similarly, the authors state that identifying the causative agent of respiratory illness is important for clinical decision making without stating what treatment options are available for each of the pathogens described in the study. They allude to this point again on line 110 without clearly stating what the outcomes of identifying the pathogen responsible are.

---

## Round 0.2 · accepted · Accept

The reviewers find that all their concerns have been addressed, it is a good manuscript.

Reviewer 2 ·

Basic reporting

no comment

Experimental design

no comment

Validity of the findings

no comment

·

Basic reporting

The authors have sufficiently addressed all previous comments. I have nothing else to add.

Experimental design

N/A

Validity of the findings

N/A

Additional comments

N/A